# Shining the Light on Astrocytic Ensembles

**DOI:** 10.3390/cells12091253

**Published:** 2023-04-26

**Authors:** Laura Delgado, Marta Navarrete

**Affiliations:** Instituto Cajal, Consejo Superior de Investigaciones Científicas (CSIC), 28002 Madrid, Spain

**Keywords:** brain integration properties, astrocytic ensembles, behavior, neuron-astrocyte communication

## Abstract

While neurons have traditionally been considered the primary players in information processing, the role of astrocytes in this mechanism has largely been overlooked due to experimental constraints. In this review, we propose that astrocytic ensembles are active working groups that contribute significantly to animal conduct and suggest that studying the maps of these ensembles in conjunction with neurons is crucial for a more comprehensive understanding of behavior. We also discuss available methods for studying astrocytes and argue that these ensembles, complementarily with neurons, code and integrate complex behaviors, potentially specializing in concrete functions.

## 1. Introduction

The complex brain structure composed of neurons, astrocytes, and the remaining glial cells has been revealed to cooperate, share information, and code responses. This communication is accomplished by using different languages, such as flows of charge across neurons that lead to electrical potentials and ionic currents, oscillatory rhythms that direct synchronization and phasing, astrocytic and neuronal calcium transients, and the continuous chemical exchange of neurotransmitters, in which all involved parties find a common language to understand. Recently, research has been shedding light on the behavior of astrocyte networks as communicating systems that have their own rules and mechanisms [1,2,3].

Astrocytes play an important role in brain function as computing hubs. Their coding strategy relies on intracellular Ca^2+^ fluctuations that are very heterogeneous in space and time [4] that result from Ca^2+^ stores released mainly by inositol triphosphate (IP_3_) receptors located in the membrane of the endoplasmic reticulum (ER) [5,6]. These signals can range from single-cell spontaneous intrinsic Ca^2+^ variations that maintain different levels of activity [7] to intercellular Ca^2+^ flows that spread over long distances and can reach hundreds of cells [8,9] to transmembrane receptors-elicited Ca^2+^ signals that account for external stimuli [5,10]. The external executors are, substantially, neurotransmitters released synaptically. Bidirectionally, astrocytes can also release neuroactive molecules, such as glutamate, ATP, GABA, adenosine, or D-serine, that switch on neuronal receptors, establishing communication and revealing their involvement in the modulation of neuronal activity and synaptic transmission in different brain areas [11,12,13,14,15,16].

Therefore, astrocytes are significantly involved in synapse formation, remodeling, and plasticity [17,18,19], which is partly due to their evolutionarily conserved intricate branched morphology—proven both in mice and humans—[20,21]. The perisynaptic astrocytic processes (PAPs) that establish physical contact with pre- and post-synaptic terminals and form tripartite synapses [22] can undergo both gross and fine-scale structural changes analogous to those seen in neurons during plasticity modulation [23,24,25,26]. This has a functional meaning since not all synapses exhibit the presence of astrocytic processes, and it is a necessity dynamically shaped under development requirements, or in response to injury, among others [27,28,29,30]. As such, astrocytes play a key role in regulating learning and memory, which are foundational to complex behaviors. To identify the astrocytic elements involved in synaptic plasticity, studies have used electrophysiology, molecular genetics, and astrocytic knockout phenotypes, as well as examining determinant elements such as receptors (e.g., cannabinoid receptor CB_1_ [31], inositol trisphosphate receptor IP_3_Rs [32,33], GABA_b_ [34], kinases [35]), cell adhesion molecules (e.g., connexins 43 and 30 [36], or neuroligins [37]), among other approaches [38,39].

In addition, astrocytes play an important role in intercellular information transmission through neuronal oscillations, which contribute to rhythmicity. They are essential participants in understanding the synchronous activities of central pattern generators (CPGs), which control behaviors such as respiration [40,41,42,43,44], locomotion [45,46,47,48,49], mastication [50,51,52], and the regulation of sleep patterns [53]. Astrocytes in the suprachiasmatic nucleus of the hypothalamus have been shown to independently encode circadian information and control the period of circadian oscillations through glutamate [54,55,56], and the inhibition of vesicular release in hippocampal astrocytes provokes disruption of gamma oscillatory activity [57]. In conclusion, the active role astrocytes play in neurotransmission is responsible for modulating the connectivity of neurons at the network level.

Therefore, despite the undeniably growing attention for the astrocytic field within the last several years (see Figure 1 and Figure 2 and Table 1), awareness of the existence and coordination of astrocyte working groups and their interaction with neuronal systems for behavioral encoding is still missing. Thus, whereas available studies generally focus on cause-effect astrocyte-neuron interactions or astrocyte function, almost none includes the panorama of network-interacting astrocytes with the view put in behavior. As a result, we detect a pivotal need to shape the concept of *astrocytic ensembles*, we discuss its supporting evidence and the conceptual relevance for implementing top-down study perspectives that ultimately allow functional correlates of astrocyte responses. Herein, we first provide an overview of the state-of-the-art methods available for studying astrocytes, showing the differences between bottom-up approaches (which are safe bets to build organized knowledge for deciphering the astrocyte’s role in the brain) and top-down perspectives (by setting the focus on astrocyte network dynamics). Next, from a systems neuroscience perspective, we discuss the features of astrocytes’ and neurons’ communication in the physiological brain as well as their connectivity patterns and argue that astrocytes’ response in behavior results from working *ensembles*. From that, we deduce that astrocytes may have the possibility to specialize, and we present current missing links to decipher astrocyte integration properties in physiological conditions. Lastly, we evaluate the most important concerns for the in vivo study of astrocytes.

## 2. Available Methods for Tracking Astrocytes

To reliably picture a system in continuous motion is an ambitious task. Herein, we review the methods available to study astrocytes and present the bottom-up and top-down study perspectives (see Table 2 and Table 3). The bottom-up approach integrates smaller pieces of information to form the bigger picture, while the top-down approach first identifies the bigger picture and then infers the relationships among the elements involved. The goal is to provide sufficient knowledge of the state-of-the-art techniques for visualizing and manipulating astrocytes to help researchers choose the right approach for their biological questions. Greater detail is beyond our scope and can be found elsewhere [6,58,59,60,61].

*Structure, morphology, and diversity.* The specialized and polarized end-foot structures of astrocytic processes vary in size, shape, and structure and require new labeling and imaging systems to accurately detect calcium transients in these fine processes [62]. This heterogeneity level leads to classification according to the diffraction-limited resolution required for its detection and, therefore, to their potential to be resolved using traditional microscopes. Discerning between adjacent astrocytes at single-cell resolution can be achieved with cell labeling and confocal microscopy by using lucifer yellow intracellular dye-filling through a sharp electrode; iontophoresis-based (which requires short tissue exposition to a fixative solution [63]); by expression of fluorescent reporters through virus delivery [64], transgenic mice lines [65], but also with electroporation [37]; or the sparse labeling method mosaicism with repeat frameshift (MORF) [66]. In addition, electron microscopy (EM) reconstructions also allow the detail of PAPs and end feet [67,68,69,70,71,72] but provide limited experimental flexibility due to requiring fixed tissue. By combining viral vector transduction to mark different cells with different fluorescent reporters, confocal or two-photon imaging, and the CLARITY technique, Goshen and Rafaeli [73] showed different astrocytes, their complete and detailed structure, and their spatial network interactions, resulting in a stationary way to represent large-scale volumes at single-scale resolutions. Spatio-temporal interactions between astrocyte processes and synapses have been detected with fluorescent proteins and a genetically targeted neuron–astrocyte assay (NAPA) that depends on resonance energy transfer (FRET) between the fluorophores expressed in neurons and the adjacent astrocytes [30,74].

Among the shaping factors of individual-cell structural and morphological diversities are genetic programs that give rise to determined molecular profiles [75]. The choice for resolving astrocytic regional and microenvironmental diversity are transcriptomic and proteomic analyses along with morphological reconstruction, which also allows for studying the occurring changes dependent on synapses, behavior, or experience [65,76,77,78]. Of concern is, therefore, the traditional limitation of molecular approaches to study transcriptomes and proteomes: the need to isolate cells (i.e., to undergo a purification process), which has been shown to change naturally occurring gene expression and does not allow spatial traceability. There are approaches to sequence RNA (RNA-seq), also single-cell (scRNA-seq) [79], including fluorescence-activated cell sorting (FACS) [80], immunopanning [81,82], and magnetic-activated cell sorting [83,84]. Translating ribosome affinity purification (TRAP) [85,86] does not require physical cell dissociation; however, the specific technique that allows for in vivo genetic targeting of selected cell populations is the protocol RiboTag [87], allowing labeling even the translatomes of subcellular compartments such as the end feet, and low-expressing genes [88], overcoming the need to having segregated cells. In regard to techniques that allow spatial transcriptomics [89,90,91], some already commercialized are Spatial Transcriptomics [92], GeoMx [93], and CosMx [94].

*Indicators of Ca^2+^ activity.* The spectrum of study possibilities commences with calcium-specific dyes (i.e., fluorescent reporters conjugated to a Ca^2+^ chelator such as Fluo-4, Rod-2-, or OregonGreen-BAPTA, among others [59,60]), which were the first used for measuring intracellular calcium [95,96]. In addition, there are genetically encoded Ca^2+^ indicators (GECIs) that bind to calcium and emit fluorescence (e.g., GCaMP [97]). Cre-dependent transgenic mice [98,99,100] and adeno-associated viruses (AAV) are available to express GECIs specifically in astrocytes by tackling specific astrocytic reporters (such as GFAP, S100β, GLAST, or GLT-1 [58,101,102]), with which it is possible to identify calcium signals across different areas, or within fine leaflets [103,104]. The combination of GECIs and optic fiber is a widely used approach to register deep-tissue measurements of calcium transients in freely behaving mice [59,103,105]. In regard to imaging, for instance, Georgiou et al. [106] used axoastrocytic AAV transfer and two-photon microscopy to study the functional-dependent organization of Ca^2+^ microdomains and were able to determine stability within occurring patterns. Additionally, in an in vivo study carried out by Li et al. [107], a combination of magnetic resonance imaging (MRI) with an AAV vector carrying a reporter was used to uncover dynamic Ca^2+^ changes in astrocytes. Furthermore, in our laboratory, we have recently adapted a functional tool to monitor astrocyte activity called CaMPARI_GFAP_, which is a GECI for astrocytes that undergoes irreversible fluorescence conversion when there is elevated intracellular Ca^2+^ and simultaneous delivery of 405 nm light [3]. Apart from allowing the study in a temporally controlled manner, it enables the labeling of a specific set of astrocytes that are active during a certain time window resulting in functional maps of the instantaneous picture of astrocyte activity. Indicators can also be genetically targeted at different signaling molecules such as second messengers or metabolites (i.e., ATP, serotonin, glucose, or glutamate, among others) to follow up the different input/output activities. Even though the simultaneous study of several molecules using different indicators can result in being restricted due to genetic constraints or fluorophores overlap, having the possibility of measuring intracellular signaling enables establishing behavioral correlates or gene expression changes and sets a step forward towards delineating the underlying computing principles.

*Actuators.* To interact with the system in an inhibitory or excitatory manner for studying the effects is an interesting approach for establishing connections between animal behavior, neural functioning, and astrocyte activity, despite the absence of strict physiological context [108]. Activation of astrocytes is traditionally performed with opto- and chemogenetic approaches such as designer receptors exclusively activated with designer drugs (DREADDs) [109,110], channelrhodopsin 2 (ChR2) [111], the G_q_-dependent actuator melanopsin [112], the light-gated ionotropic glutamate receptor LiGluR [30,113], or the engineered light-sensitive GPCRs called opto-XRs [114]. As for inhibition, multiple agonists tackling different targets can be used depending on the source of the input signal (e.g., signals mediated by G_q_ GPCR [115] or Ca^2+^ release from intracellular stores in which IP_3_ type 2 receptors are involved [116]). With calcium extruders, which is a genetic approach, it is also possible to attenuate calcium signaling regardless of the source [117].

*Advanced* in vivo *imaging technologies.* Emerging imaging technologies are two-photon laser scanning microscopy (2PLSM)—which reaches significant tissue depths in the order of 500 nm—[118,119]; laser scanning confocal microscopy (LSCM)—suitable for more superficial areas—[120]; stimulation emission depletion (STED)—a right choice for fine process detection—[121]; or light sheet fluorescent microscopy (LSFM)—that allows acquisition of ultrafast calcium transients—[122,123]. For instance, using 2PLSM, Volterra et al. [124] tracked in 3D the Ca^2+^ activity of astrocytic processes in the hippocampus, revealing different types of events and correlating them with local synapses. Nevertheless, methods such as two-photon stimulation emission depletion (2P-STED) are emerging for in vivo exploration of interactions under physiological conditions, allowing correlation with activity dependence.

*Analysis tools.* Ultimately, the efficacy of manipulating and imaging methods will also depend on the development of analytical tools [58,60] to focus the resulting data toward predictive modeling. Available software for analyzing Ca ^2+^ signals includes Astrocyte Quantitative Analysis (AQuA) [125], which is a non-ROIs (regions-of-interest)-based analysis that captures the heterogeneous fluorescent dynamics, allows spatio-temporal quantification and is a good choice for examining dynamical properties of distinct calcium transients. In contrast, GECIquant [126] determines ROIs containing calcium changes based on a certain threshold value, permits differentiating between process-occurring activity or somatic fluctuations, and is better suited for examining localized calcium transients within single cells. An ROI-based machine learning method is CaSCaDe [127], which outputs a spatial map of activity and can be used to monitor single-cell calcium microdomains. Partition in Regular Quadrants (PRQ), which is ROI-based, also enables a spatial analysis to obtain activity maps [3] or the MATLAB custom program Calsee [128] to contour the territories of individual astrocytes and quantify region-specific calcium activity. The choice between ROI- or event-based methods should be made based on how limiting it is that ROIs are constant in shape, position, and overlap [125]. In terms of network analysis, Astral was designed to analyze astrocyte–astrocyte population interactions [129] and provides an event-based approach to measure single-cell events and their intercellular propagation.

To summarize, experimental top-down study methods in astrocytes have been very limited until recently, which has considerably slowed down the possibility of setting the focus on astrocyte network dynamics. However, now there are available techniques to combine the in vivo labeling of active astrocytic ensembles with quantitative measures of their activity (with advanced imaging or fiber photometry recordings). This enables tracking the astrocytic activity at the network level during a certain behavior but also deciphering the astroglial physiological activation patterns, which are essential for establishing relationships to infer the integration properties of astrocytes.

**Table 2 cells-12-01253-t002:** Experimental techniques to label and visualize astrocytes.

Experimental Techniques		
Structure and/or Morphology of Astrocytes		References
**Labeling**	Lucifer yellow intracellular dye-filling through sharp electrode, iontophoresis based	[63]
	Expression of fluorescent reporters through virus delivery	[64]
	Transgenic mice lines	[65]
	Electroporation for virus delivery	[37]
	Sparse labeling method mosaicism with repeat frameshift (MORF)	[66]
**Imaging**	Electron microscopy reconstructions	[67,68,69,70,71,72]
	Two-photon laser scanning microscopy (2PLSM)	[118,119]
	Laser scanning confocal microscopy (LSCM)	[120]
	Stimulation emission depletion (STED)	[121]
	Light sheet fluorescent microscopy (LSFM)	[122,123]
**Labeling and Imaging**	CLARITY	[73]
**Labeling and Imaging and FRET Phenomenon**	NAPA	[30,74]
**Genetic Programs and Molecular Profiles**		
**RNA-Seq and Single Cell RNA-Seq**	Fluorescence-activated cell sorting (FACS)	[80]
	Immunopanning	[81,82]
	Magnetic-activated cell sorting	[83,84]
	Translating ribosome affinity purification (TRAP)	[85,86]
	RiboTag	[87,88]
**Spatial Transcriptomics**	Spatial Transcriptomics	[92]
	GeoMx	[93]
	CosMx	[94]
**Activity of Astrocytes**		
**Ca^2+^ Specific Dyes for Intracellular Measurements**	Ca^2+^ indicators (Fluo4, Rod2, OregonGreen-BAPTA)	[59,60]
**Genetically Encoded Ca^2+^** **Indicators** **(GECIs)**	GCamp	[3,97]
	Cre-dependent transgenic mice	[98,99,100]
	Adeno-associated viruses	[58,101,102,106,107]
**Activation of Astrocytes**	Dreadds	[109,110]
	ChR2	[111]
	Melanopsin	[130]
	Light-gated ionotropic glutamate receptor LiGluR	[30,113]
	GPCRs opto-XRSs	[114]
**Inhibition of astrocytes**	Pharmacological agonists	[115,116]
	Calcium extruders	[117]

**Table 3 cells-12-01253-t003:** Analytical tools to study astrocytes.

Analytical Tools	References	Properties
Astrocyte Quantitative Analysis (AQuA)	[125]	Non-ROIs based Spatiotemporal quantification of fluorescent dynamics Dynamical study of Ca^2+^ transients
GECIquant	[126]	ROIs based Discerns between process or somatic occurring activity Single cells
CaSCaDe	[127]	ROIs based Spatial maps of activity To monitor single-cell Ca^2+^ microdomains
Partition in Regular Quadrants (PRQ)	[3]	ROIs based Spatial maps of activity
Calsee	[128]	To delineate individual astrocytes and quantify activity region-specifically
Astral	[129]	To analyze network population interactions among astrocytes To measure single-cell events and intercellular propagation

## 3. Astrocytes Add Complexity and Increase Integration Versatility

### 3.1. Definition of Functional Astrocytic Ensembles

The central axis of systems neuroscience for understanding the biological complexity of the brain is to seek a systems-level description of the network interactions regardless of analyzing its individual components. To date, the discipline barely considers the role of glial cells [131]. In the following lines, we focus on the interactions between neuronal and astrocytic systems and on their different connectivity patterns, shaped by their distinct electrical, structural, and temporal features.

Neurons and astrocytes have developed their own individual attributes that lead to different network working principles. On the one hand, neurons have high electrical excitability, while astrocytes have the capacity for K^+^ buffering and, therefore, for controlling this electrical excitability of neurons. This difference in electrical barriers may allow the overall system to balance activation and achieve maximum efficiency. On the other hand, astrocytes form a continuous assembly due to their connection through gap junctions that facilitate intercellular coupling, control the flow of molecules, and allow the sharing of cytoplasm, leading to a reticular organization known as syncytium [132], while neurons form a network of independent connected units. The divergence in electrical and structural features between astrocytes and neurons is also found in the temporal resolution dynamics that provide each system with a separate, complementary field of action [31,124,133].

The modularity theory is well-assumed in most theoretical approaches in cognitive science to represent innate neural structures that work together to form functional circuits [134,135]. In fact, there are supportive studies that report, for instance, how (1) by externally triggering certain neuronal patterns, it is possible to induce motor tasks in humans [136], (2) by triggering the electrical encoding of facial identity, it is possible to reproduce it with a high degree of accuracy in macaques [137], or even (3) decision-making can be predictable from action potential sequences with a wise choice probability above 70% [138,139]. Nevertheless, none include the response refinement encoded by astrocytes, missing an important variable that should be considered indispensable [13,34,54,58,108,115,140,141,142,143].

There are no solid responses regarding how astrocytes encode outputs to influence behavior, despite the evidence showing their involvement. An example is found in the astrocytic implications of memory consolidation, a complex behavior that requires the integration of reasoning, comprehension, and learning in the case of humans. In 2012, Han et al. [31] showed that activation of astroglial cannabinoid receptors is sufficient to impair spatial working memory (SWM) and induce long-term depression (LTD). Recently, a study published by Becker et al. [133] showed how working memory representations are influenced by astrocyte signaling by developing a computational model of a neural network that considers astrocytes, which resulted in actions on presynaptic short-term plasticity (STP) dynamics by increasing its calcium binding rate. Namely, they show how the fast STP dynamics are modulated by the slow timescale of astrocytes that ranges from seconds to minutes suggesting a slow integrator role that also saves information about the active representation. Gordleeva et al. [144] go further in the modeling by considering astrocyte–astrocyte interactions and draw conclusions on their ability to store patterns to coherently perform synaptic changes, incorporating novel biophysical network mechanisms of memory formation. The latter is supported as well, among others [145,146,147], by Porto-Pazos et al. [148], who concluded that artificial neuron–glia networks (NGN) provide more efficient performance in comparison to sole artificial neural networks (NN) and that this increased complexity is a factor for network performance improvement.

Therefore, the conclusion that astrocytes are indispensable in brain function related to behavior is emerging in parallel to the growing evidence showing how astrocytes are organized into functional clusters. In the research carried out by Houades et al. [149], it is shown that astrocytes in the barrel cortex couple largely through gap junctions in the same barrel and scarcely between astrocytes from different barrels, which define local microcircuits. Apparently, the established connectivity is region-specific, revealing a non-homogeneous character of gap junctions and maybe interacting astrocytic subpopulations. Sul et al. [150] concluded that adjoining astrocytes in the hippocampus were functionally connected and that this recruitment was determined by certain threshold concentrations of the extracellular transmitters ATP and ACh. However, how this specificity is modulated remains unknown. Furthermore, in a recently published study by our laboratory by Serra et al. [3], the stimulation of three different glutamatergic nuclei resulted in differential astrocytic activity in the nucleus accumbens in a way that for each stimulated nucleus astrocytes not only responded in highly innervated areas, which supports the existence of different connected astrocyte subpopulations able to discern the input origin. This astrocytic discrimination between pathways was also previously observed by Perea and Araque [151]. Moreover, the encoding of spatial information by hippocampal astrocytes has been experimentally investigated by Curreli and colleagues [152], who conclude that the coding is not explained by mere visual cue information. They give evidence about how spatial information was processed differentially in distinct topographical locations of the same astrocyte and that it complemented neuron-encoded information in an excluding way. Doron et al. [2] enriched Curreli’s findings by tracking real-time calcium activity of astrocytes in the hippocampus also during a spatial event-related task. The results showed how astrocytic calcium activity increased after recognition of a learned reward location. The latter means that astrocytes code information levels since the increase was not appreciated in non-familiar environments. Importantly, they showed that the reconstruction of a mouse trajectory was feasible only with the activity of the astrocytic population.

Consequently, by combining the theoretical framework of the modular theory with the experimental findings that support the existence of subsets of astrocytes that activate in a behavioral paradigm, we detect there is a missing concept to refer to these groups of working astrocytes and, herein, we propose the term *astrocytic ensembles* (see Figure 3). This conceptualization will aid the phenomenon in being formulated into testable hypotheses and correlated with specific variables of interest. Lastly, an agreement in the understanding of such a key concept opens up a horizon of questions about the mechanism behind ensemble selection and whether it happens in all behavioral conditions, how the behavioral ensembles operate in an unconditioned neutral context, or whether gradients of positive or negative valences are encoded. The experimental approaches to these responses implicate the selective labeling of implicated ensembles and their reactivation or inhibition.

### 3.2. Could Astrocytes Undergo Specialization?

The study of astrocytes is conditioned by their heterogeneous nature [75,153,154,155]. Initially, two main astrocyte types were characterized, the fibrous and protoplasmic, found in white and gray matter, respectively [156]. However, currently, there is evidence showing the existence of several phenotypes with determined molecular profiles that differentially express ion channels [157], gap junction proteins [158], glutamate transporters [159,160], or receptors [161]. The individual cell identities are shaped by intrinsic factors (developmental patterning that results in a unique combination of transcription factors [156,162,163]) and extrinsic factors (the local environment drives functional adaptations, i.e., the mature phenotype is highly determined by the signaling from adjacent neurons, despite coming from the same progenitor pool [65,164]). What is more, the different phenotypes have been shown to be located according to the brain region’s functionality [65,78,163,165]. Therefore, astroglial cells exhibit high heterogeneity [154,155], not only in their morphology [65,75] and developmental origin [156,162,163] but also in their gene expression profiles [18,75,153] and functionalities [6,141,154,165], both within the same region and across different areas [65]. While the functional implications of this diversity in various neural circuits is a current matter of debate, astrocytes play critical roles in the formation, maturation, and maintenance of synapses in diverse brain regions [14,22,25], both in physiological and pathological conditions [75,166]. Astroglia are constantly facing changing environments, and this environmental degree of complexity increases throughout the phylogenetic scale. In this way, compared to rodent astrocytes, human astrocytes are larger, structurally more complex, and show a higher phenotypic diversity, and their Ca^2+^ signaling travels at larger velocities [167]. These features provide the human brain with increased functional competence and more complex computational mechanisms [167,168,169].

In the case of neurons, it is well known that they are specialized cells with defined functions [131,170], implying that they have well-conserved roles and can perform the same role in different physiological situations. By contrast, astrocytes show very variable capacities, which permits them to perform very different roles in different physiological situations. However, under the assumption that the difference between heterogeneity and specialization lies in the attribution of a functional role, we deduce that astrocytic ensembles are likely to be specialized for a determined function. In this line, how could, therefore, the dynamic nature of astrocytes’ cell identity be explained? Do ensembles lose their capacity to operate with such a margin of versatility? How is specialization controlled, and when is it triggered? Contemplating the limitations for capturing relevant physiological sub-populations of astrocytes and exploring alternative parameters to define functionally homogeneous populations to study their impact on neural circuits reveals the ever-evolving nature of neuroscience research.

### 3.3. The In Vivo Study of Astrocytic Processing in Behavior

There is no doubt that astrocytes play important roles in the production of cognition, emotion, sensory, and motor processing that give rise to complex behaviors [140,171]. The current knowledge of astrocytic calcium signaling mainly comes from studies of responses to specific ex vivo or in vitro stimulation of different brain regions [154], but in order to understand the correlation between behavior and astrocytic activity, in vivo studies in awake animals are requisite. Until now, the main limitation for attempting the in vivo tracking of astroglial Ca^2+^ activity has been the lack of available methods, and most studies were based on anesthetized animals [105], which have been useful to determine some encoded patterns of the processing strategies but leave unattended the physiological condition.

Astrocytes activate through intracellular calcium increases, which are their basic unit for communication [4,15]. Multiple parameters of the signaling encode information, such as the amplitude, duration, frequency, transient onset, or spatial pattern [172,173], which provides the astrocytes with a very high versatility of operation. For instance, studies in anesthetized mice have shown that astrocytes in the somatosensory cortex respond to foot stimulation by modulating their signal amplitude [174] or that the frequency of intense whisker stimulation is encoded in the number of recruited astrocytes in the barrel cortex [175]. However, general anesthesia has been shown to have a significant effect on calcium signaling, which can affect calcium transients and spontaneous activity [176,177,178]. To overcome these limitations, studies using GECIs and fiber optic recordings have been conducted in freely moving animals, bringing the challenge of interpreting the signals, as the recordings contain information from non-relevant sources. The fact that astrocytic activity could be masked in the awake in vivo state has been addressed by Curreli et al. [152]. They demonstrated that the signaling of hippocampal astrocytes encodes information regarding the animal position and that this information is complementary to that of neural place cells. Additionally, Doron et al. showed that hippocampal astrocytes ciphered a reward location only when the spatial context was familiar. Even more, they demonstrated that the activity of the astroglial population was sufficient to reconstruct trajectories in known environments [2]. In summary, there is evidence showing how astrocytic signals encode information complementarily to neurons and support their involvement in the refinement mechanisms of neuronal responses with integrative (and stabilizing) properties [12].

Therefore, these studies suggest that the existence of astrocytic networks enables complex integration in the cognitive function of spatial information coding, including the processing of episodic memory linked to specific locations and contexts. Importantly, these results reflect physiological astrocytic responses. Additionally, the ability of astrocytes to reflect the activity of neighboring neurons [175,178] and encode complementary information [152] may depend on the type of response elicited, either sensory-evoked or event-based. Further research is needed to better understand these differences and to monitor specific cell identities in specific networks for different tasks.

The potential to comprehend the computational principles of astrocytes is now within reach. Can we anticipate a uniform calculation between inputs and outputs within the same framework? How would the stimulation of astrocytes specifically impact the dynamics of neuronal efferents? Is there a noticeable behavioral effect after stimulating astrocytic ensembles in vivo? In light of the studies discussed in Section 3.1, where specific neuronal networks were selectively stimulated, resulting in a motor outcome [137,138,139], we hypothesize that triggering the selective activation of astrocytic ensembles could imply or enhance a behavioral circumstance, probably in the scope of a learned task due to the refinement and integrative character of their responses. For this, it is remarkable that the necessity of having new tools allows the tracking of the Ca^2+^ levels of astrocytic activities.

## 4. Concluding Remarks

Astrocytes are active cells implicated in the computational procedures of the brain. Based on recent studies showing that astrocytes work as networked systems and the modular nature of the brain, we propose the term “ensemble” to describe groups of astrocytes that come together to carry out a specific function. This concept opens up the possibility that these ensembles may develop specialized skills for performing that function. By making a systematic comparison of neurons and astrocytes, our argument rotates around the requirement of increasing the number of experimental studies that consider the existence of astrocytic ensembles since the number of currently available ones is very limited and computational descriptions need to be based on or supported by experimental facts. Despite the well-known fact that astrocytes play a role in animal behavior—their activity disruption being a cause of disease—how the coding of their responses occurs is not yet determined.

It is important to emphasize that astrocytes interact with other glial cells, including microglia [179,180,181] and oligodendrocytes [182,183], and also with immune cells [184,185]. These interactions play a crucial role in both normal and disease conditions [185,186,187]. Recent experimental evidence has confirmed the significant role of astrocyte-microglia communication [186], which has therapeutic potential for neurological disorders such as experimental autoimmune encephalomyelitis (EAE) and multiple sclerosis (MS) [188,189]. Therefore, studying the mechanisms of communication within astrocytic engrams and other glial cells will help us understand disease-promoting glial responses and develop new treatments. Similarly, the finding that astrocytes can be reprogrammed into neuronal progenitors and mature neurons in the CNS of adult mammals [190,191,192,193,194,195] raises questions about the role that ensembles may play in this orchestration. The molecular mechanisms, regulatory networks, and potential astrocytic ensembles that trigger this reprogramming are not yet clear, which could bring new insights into regeneration-based therapies.

The fact that we currently have methods to study astrocytic ensembles in mice and rats is not yet promising in the study of the human brain astrocyte ensembles due to their being highly invasive. It is, however, worth mentioning that some of the studies herein referenced were performed in humans or in non-human primates [104] due to their being non-invasive [68,136] or were with the informed consent of patients in surgeries of epilepsy cases and tumors [82].

The task of deciphering the results generated by neuronal–astrocyte networks is complex, involving the systematization of mechanisms or the creation of approximate fitting models. By adopting a top-down perspective and concentrating on the dynamics of astrocyte networks, it is possible to make headway in identifying the functional correlates of astrocyte responses. Such insights will establish the groundwork for more sophisticated investigations and contribute towards comprehending potential therapeutic targets aimed at repairing and restoring normal function in disease.

## Figures and Tables

**Figure 1 cells-12-01253-f001:**
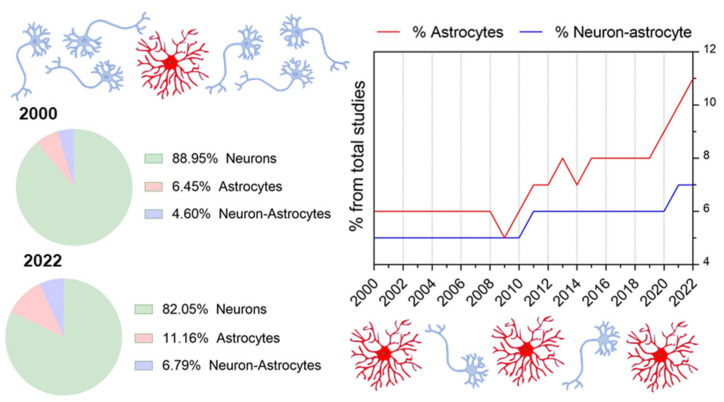
Percentage of studies focusing exclusively on Neurons (green), Astrocytes (red), and on Neurons and Astrocytes (blue) published in the years 2000 and 2022 and across time within the year range from 2000 to 2022. The total number accounts for the sum of Neuron, Astrocytes, and Neuron and Astrocytes studies. Data were extracted from the Scopus database.

**Figure 2 cells-12-01253-f002:**
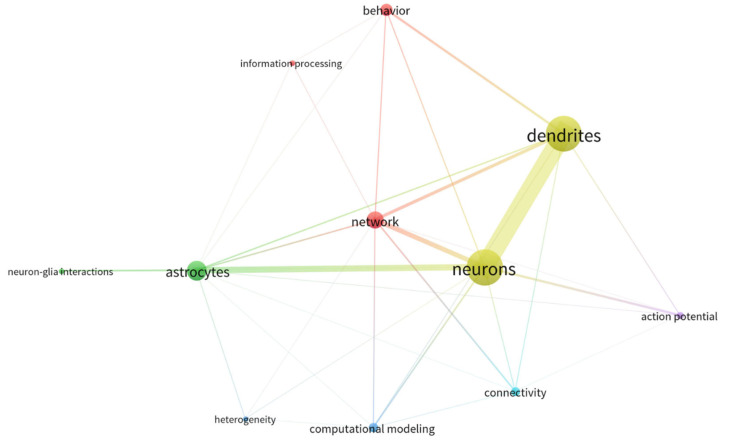
Map of bibliographical data with co-occurrence analysis of arbitrarily selected keywords (items) to show the relatedness of the keywords as a function of the number of documents in which they occur together, which is quantified by a weight value, and visually reflected in the thickness of the linking line, the closeness within items, and the label size.

**Figure 3 cells-12-01253-f003:**
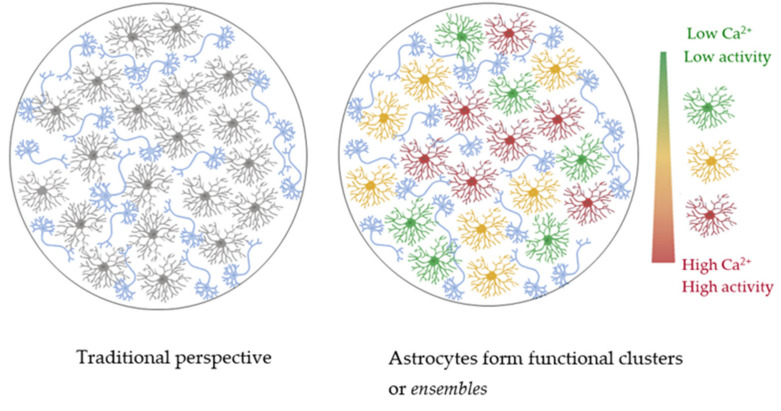
Schematic representation to illustrate the traditional perspective in which astrocytes respond homogeneously (**left**; neurons (blue) and astrocytes (gray)) and the view in which astrocytes respond in ensembles (**right**; neurons (blue), low-activity astrocytes (green), medium-activity astrocytes (yellow), high-activity astrocytes (red)).

**Table 1 cells-12-01253-t001:** Details of the bibliographical search seen in Figure 1.

Bibliographic Search	
Source	Scopus
Data accessed	26 October 2022
**Query**	
Description	(neuron) OR (astrocyte) AND (mice, OR mouse, OR human) AND (dendrite) AND (process) AND (network, OR engram)
Documents found	16.199
**Query Filters**	
Years Range	1980–2023
Language	English
**Map Analysis**	
Software	1.6.18 VOSviewer
Type	Co-occurrence
Counting Method	Full counting

## Data Availability

Not applicable.

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
