# Peer review of "Shining the Light on Astrocytic Ensembles"

_cells, 2023, doi:10.3390/cells12091253_

Round 1

Reviewer 1 Report

In this manuscript, the authors first provide a comprehensive introduction to the research methods of astrocytes, and then focus on discussing the diversity and functionality of astrocyte ensembles. However, before it can be accepted for publication, a few issues need to be addressed.

(1) In Section 2, it is recommended to add a table summarizing the relevant methods to improve the readability of the manuscript.

(2) The heterogeneity of astrocytes is currently one of the research focuses in the field. We recommend adding a paragraph in Section 3.2 summarizing the current understanding of heterogeneity of astrocytes.

(3) In the last paragraph of Section 3.3, the authors raised several key questions about astrocyte ensembles. We suggest that the authors also provide some perspectives after presenting these questions.

(4) The discovery of astrocytes reprogramming into neurons is an exciting advancement in the field, which demonstrates astrocytes have the potential to supplement or replace neurons under certain physiological or pathological conditions. We recommend adding a discussion on this aspect.

(5) Astrocytes have close interactions with many types of cells. Recently, Wheeler et al. (Science 2023) demonstrated the cross-talk between astrocytes and microglia, which is an important new function of "astrocytic ensembles". We recommend adding a discussion on this finding.

(6) Different colored cells in Figures 1 and 3 should be labeled in the legends.

Reviewer 2 Report

1. This review brings into focus recent work examining the coordinated activities of groups of astrocytes, which the authors call “ensembles.” The Introduction sets out much of what is known about astrocyte function, although this section will not be of much interest to glial biologists, since it is a statement of what is already well known.  It can be shortened. 

2. The review gives an extensive list of references.  This is excellent for those entering the astrocyte field or glial biologists who want to expand their studies.

3. In Section 2, the available methods is simply a listing of papers that use certain techniques.  This could be useful to investigators beginning to study astrocytes, although for most of these methods the authors do not give the reader any sense of potential positive value or difficulties. The authors’ opinions would help those who wish to get into this field. The discussion of Ca++ imaging techniques is excellent, not surprisingly since the authors are experts in that area.

4. The review almost entirely cites papers that work with mouse CNS.  There is of course much less work done on human and non-human primate astrocytes, and some of those studies are cited.  Perhaps the authors should either note when they are talking about primate astrocytes or put in a paragraph about primate astrocytes.

5. The review is fluently written, although there are grammatical changes that should be made.  Here are a few examples, listed by line, with suggestions.

25 have their own rules

40 “communication” not “talking”

60 this is not grammatically correct

74 “astrocytes”, not “astrocyte’s”

77 on the existence

80 astrocyte, not astrocytes

80 almost none includes

81 with the view put in behavior – not clear what is meant here

88 networks’

217 “analytical” not “analysis”

248 “glial” cells

251 developed their own

261 each system with complementary but separate fields of action

270 none includes

273 “evidence”, not “evidences”

280 which resulted in actions on presynaptic

295 which define local microcircuits

300 ACh

304 astrocytes responded only in highly

342 by their heterogeneous nature

380 “which”, not “what”
